# Clinical and Morphological Aspects of Aggressive Salivary Gland Mixed Tumors: A Narrative Review

**DOI:** 10.3390/diagnostics14171942

**Published:** 2024-09-03

**Authors:** Alexandra Corina Faur, Alina Maria Șișu, Laura Andreea Ghenciu, Roxana Iacob, Emil Robert Stoicescu, Ovidiu Alin Hațegan, Mărioara Cornianu

**Affiliations:** 1Department of Anatomy and Embryology, “Victor Babeș” University of Medicine and Pharmacy Timișoara, Eftimie Murgu Square, No. 2, 300041 Timișoara, Romania; faur.alexandra@umft.ro (A.C.F.); roxana.iacob@umft.ro (R.I.); 2Department of Functional Sciences, “Victor Babeș” University of Medicine and Pharmacy Timișoara, Eftimie Murgu Square, No. 2, 300041 Timișoara, Romania; bolintineanu.laura@umft.ro; 3Doctoral School, “Victor Babes” University of Medicine and Pharmacy Timisoara, Eftimie Murgu Square No. 2, 300041 Timisoara, Romania; 4Department of Radiology and Medical Imaging, “Victor Babeș” University of Medicine and Pharmacy Timisoara, Eftimie Murgu Square No. 2, 300041 Timisoara, Romania; stoicescu.emil@umft.ro; 5Discipline of Anatomy and Embryology, Medicine Faculty, “Vasile Goldis” Western University of Arad, Revolution Boulevard 94, 310025 Arad, Romania; hategan.ovidiu@uvvg.ro; 6Department of Microscopic-Morphology-Morphopathology, ANATPATMOL Research Center, “Victor Babeș” University of Medicine and Pharmacy Timișoara, Eftimie Murgu Square, No. 2, 300041 Timișoara, Romania; cornianu.marioara@umft.ro

**Keywords:** carcinoma ex pleomorphic adenoma, salivary carcinosarcoma, metastasizing pleomorphic adenoma, mixed tumors of the salivary glands, minor salivary glands, salivary tumors

## Abstract

Salivary gland tumors are a rare and heterogeneous group of neoplasms of the head and neck region. The mixed category of these tumors include the following entities: pleomorphic adenoma (PA), carcinoma ex pleomorphic adenoma (CEPA), salivary carcinosarcoma (CS), and metastasizing PA (MPA). The most common benign tumor of the salivary glands is PA. Metastasis and malignant degeneration have been reported in cases of PA of a salivary gland origin. Judging by their behavior, MPA, CEPA, and CS can be considered aggressive tumors. Invasive CEPA has been identified in the parotid gland more frequently. MPA and CS cases reported in the current literature are rare. In this paper, we present, narratively, the clinico-morphological features of this group of mixed tumors.

## 1. Introduction

Salivary gland tumors are a heterogeneous group of neoplasms accounting for 5% of malignant lesions in the head and neck region [1]. The entities usually included in the category of salivary gland mixed tumors are pleomorphic adenomas (PAs), carcinoma ex pleomorphic adenoma (CEPA), carcinosarcoma (CS), and metastasizing PA (MPA). PA is the most common benign tumor of the salivary glands, with an incidence rate of up to 70%. Approximately 6.2% of PAs can undergo malignant transformation. CEPA is a generic term that defines malignant epithelial tumors developed from a pre-existing PA. For a CEPA diagnosis, both carcinomatous areas and benign PA must be observed in the histopathological section or there must be a previously diagnosed PA at the respective location. CS, or a true malignant mixed tumor, is a biphasic tumor with both epithelial/myoepithelial and mesenchymal malignant elements, arising de novo or from a preexisting PA. MPA is a rare occurrence, histologically similar to PA, but produces secondary tumors in distant sites [2,3,4,5].

CEPA represents 3.6% of salivary tumors (varying between 0.9 and 14% depending on the study) and 12% of malignant salivary tumors (varying between 2.8 and 42.4%), with an incidence rate of 0.17 tumors per 1 million inhabitants [6,7,8,9]. CEPA has a slight female predilection and typically develops in patients aged 60–80 years, with a median age of 60 years, making CEPA patients on average 12 years older than those with PA [10,11]. A study focusing on CEPA in major salivary glands reports a higher frequency among male patients in their 5th decade of life, while another study indicates that invasive CEPA is more common in males over 60 years old [12,13]. CEPA is more frequently found in the parotid gland but can also be observed in the submandibular gland and, in less than 7% of cases, in the minor salivary glands of the palate, nasopharynx, oral mucosa, maxilla, nasal cavity, lower lip, and less often in the upper lip [6,10,11,14,15,16]. More unusual locations include the lacrimal glands or the accessory parotid gland [17,18]. Few studies have identified malignant salivary gland tumors in the sublingual gland, with even fewer cases of CEPA in this location [19,20]. Some authors report that aggressive variants of CEPA are more frequently located in the parotid gland (81.7%), with 18% of cases in the submandibular gland, 0.3% in the sublingual gland, and some cases affecting minor salivary glands, such as the palatine glands [13,14,21,22].

CS is a malignant salivary tumor composed of both carcinomatous and sarcomatous elements. CS, described by Kirklin in 1951, and termed true mixed malignant tumor (carcinosarcoma) by King OH Jr. in 1976 [3,23], represents less than 1% of malignant mixed tumors (0.04–1%), with only about 100 cases reported in the current literature [3,7,23,24,25]. Most patients have a history of PA or present with histopathological aspects of PAs. In rare cases (0.2%), CS arises de novo [26]. CS is more common in patients in their 6th to 7th decades of life, with a slight male preponderance and an average presentation age of 58 years (ranging from 14–87 years). About two-thirds of the tumors develop in the parotid gland, approximately 19% in the submandibular gland, and 14% in the palate. Some studies report more frequent CS occurrence in the larynx and less often in the nasopharynx, tongue, oral floor, gingivae, or major salivary glands [3,7,27].

MPA is a benign tumor that inexplicably metastasizes regionally or distantly [28]. These rare tumors, with only a few cases described, develop in the parotid gland (over 75%), the submandibular gland (13%), and the palate (9%). Some studies hypothesize that MPA and CEPA represent different stages of PA malignancy [6,29,30]. CEPA with metastatic deposits in the kidney, lung, and bone composed exclusively of MPA has been described [6]. Knight J and Rathasingham K reviewed 81 cases of MPA in the salivary glands over 72 years (1942–2014) and found that these tumors most commonly metastasize to bone, lung, and neck lymph nodes. The mean age at diagnosis is 49.5 years, with 74.1% of cases in the parotid gland, 14.8% in the submandibular gland, 6.2% in the palate, 2.5% in the nasal septum, and 1% in the tongue. The reported male-to-female ratio is 34:46, with 80.4% of patients alive at a 1-year follow-up. In their study, MPAs were identified in bone (36.6%), lung (33.8%), cervical lymph nodes (20.1%), renal (8.6%), cutaneous (8.6%), liver (4.9%), and brain (3.7%). Thirty-three of these cases had multiple sites of metastases. Isolated cases of MPA have been described in the sinus, retroperitoneal space, abdominal wall, pharynx, mediastinum, and breast [4]. This is the largest series of MPAs described in a review paper identified in the current literature. Additional studies report individual cases of MPA mainly localized to lymph nodes, with fewer cases in the lung, bone, and kidney. The distribution of MPA cases is illustrated in Figure 1 [4,30,31,32,33,34,35,36].

In this narrative review, we discuss the morphological and clinical aspects that characterize the mixed tumors of the salivary glands that may aid in the understanding of their diagnosis and the evolutionary course.

### Materials and Methods

We have chosen to prepare the review in the narrative form because of the rarity of the salivary gland neoplastic lesions reported in the literature (mostly case report articles). We consulted the PRISMA criteria and adapted the method for our study [37]. We had a purpose of identifying the articles with the subject of mixed salivary gland tumors. English-language literature covering the diagnosis of salivary gland tumors was searched by accessing the PubMed electronic database and other sources (Google scholar, Scopus). The scientific publications were hand-searched in the internet data using the following key words: “carcinoma ex pleomorphic adenoma”, “salivary carcinosarcoma”, “metastasising pleomorphic adenoma”, “mixed tumors of the salivary glands”, “minor salivary glands,” and “salivary tumors”. We identified and overviewed a database comprising 205 research papers addressing salivary gland neoplasms, comprising both individual studies and reviews. Abstracts, duplicates, irrelevant topics, publications in other languages than English, and articles not in the field of interest or with repetitive information were excluded from the study. The final number of obtained articles was 80. We categorized these remaining articles into four datasets. One dataset comprised studies investigating salivary glands tumors with an emphasis on the subject of mixed salivary neoplasms. The remaining three datasets each focused on a specific type of the mixed salivary gland tumor (CEAP, CS, and MPA). The algorithm used for this research paper selection is explained in Figure 2.

## 2. Clinico-Morphological Aspects of CEPA

### 2.1. Clinical Aspects

Clinical data considered, suggestive of the possibility of malignancy of a PA, are as follows: the submandibular location of the tumor, a longer period of time for development, older patient age (average 61 years), and tumors larger than 4 cm [2]. In cases of CEPA, two main clinical aspects are present: either the patients report the presence of a slow growing tumor mass, which is rapidly increasing in size, or the malignant evolution of a PA is noted in a patient that has underwent several surgeries [10]. However, most CEPAs, although invasive, are asymptomatic [7].

The typical history of a patient with CEPA is represented by the presence of a tumor mass for a period longer than 3 years (10–15 years), which begins to increase rapidly in a few months (on average 3–6 months). There are patients with tumors increasing in a shorter period [6,7]. Tumors arising in the minor salivary glands represent 9–23% of all salivary neoplasms. The incidence of minor salivary gland CEPA cases is difficult to determine due to their rarity [21,38,39,40,41,42]. In 30% of cases (especially those located in the major salivary glands), patients complain of pain and facial nerve paralysis, enlarged lymph nodes, dysphagia, toothache, skin ulcerations, and masses of tumors fixed to the skin. The periods for which the symptoms lead the patient to seek medical advice vary from one month to 52 years. In cases with a sinonasal location, a symptomatology (of variable duration in time from 1 to 60 months) represented by symptoms of obstruction-difficulties in breathing and chronic sinusitis, but also epistaxis, headache, tooth dislocations, or otitis media, was described [6,7,8,9]. A patient with CEPA of the minor salivary glands with lung metastasis presented with a painful ulcerative lesion of the wall of the oropharynx [21].

### 2.2. Gross Aspects

On average, CEPA is twice as large as its benign counterpart, ranging from 1.5 to 25 cm in diameter. Most of these tumors are poorly circumscribed and infiltrative, though if PA is the dominant component, they may be well circumscribed, encapsulated, or fibrotic [7,43]. Extensive sampling of the PA surgical specimen is crucial to evaluate possible CEPA, and multiple serial sections are sometimes required to identify the malignant area. Grossly, PAs in the major salivary glands are generally well circumscribed and encapsulated, while non-encapsulated tumors are more frequently identified in the minor salivary glands. The degree of encapsulation in PAs varies, with some areas lacking a capsule and showing tumor extension into adjacent tissue. However, if these areas are continuous with the PA, they should not be considered as tumor invasion. Tumor extension from the capsule is seen in recurrent or slow-growing PAs. In invasive CEPA, the carcinomatous area is often more extensive than the benign tumor. The cross-sectional surface of CEPA may exhibit any of the macroscopic features found in PAs and in the corresponding types of carcinomatous components. More commonly, bluish-grey, transparent, or whitish-yellow tumor masses with areas of calcification are described. In some cases, where the malignant component is dominant, areas of hemorrhage and necrosis are present [2,9,13].

### 2.3. Microscopic Aspects

Any type of carcinoma can be identified as the carcinomatous component of CEPA. Most commonly, the carcinomatous component is represented by adenocarcinoma not otherwise specified (42.4%) and salivary duct carcinoma (32.8%). Less frequent types include mucoepidermoid carcinoma (MEC), adenoid cystic carcinoma (ACC), undifferentiated carcinoma (UC), myoepithelial and epithelial–myoepithelial carcinoma (EMC), and polymorphous adenocarcinoma [7,13,43,44,45]. The MEC component must be graded into low-, intermediate-, and high-grade tumors, as identifying high-grade transformation areas in CEPA is crucial for predicting patient outcomes [28,46]. Figure 3 and Figure 4 illustrate examples of CEPA with salivary duct carcinoma (SDC), adenocarcinoma not otherwise specified (ADK NOS), and MEC as malignant components.

Rare occurrences of CEPA are reported in the literature, such as small-cell carcinomas developed on parotid PA or ACC developed in sinonasal PA [10,47]. Iino et al. identified a case of CEPA with a clear cell squamous cell carcinoma (SCCcc) as the carcinomatous component. Histochemical staining, including the absence of Alcian Blue-positive mucinous cells, ruled out mucoepidermoid carcinoma, making SCCcc a diagnosis of exclusion [48]. Karpowicz et al. described a CEPA with melanoma as the malignant component [24,49]. Reports of acinic cell carcinomas as the malignant component in CEPA are few [50,51].

In CEPA, the proportion of benign versus malignant components varies. Some authors categorize CEPA, based on the malignant component’s appearance, into cases with only an epithelial component and those with a myoepithelial component. Cases with exclusively myoepithelial components are rare and tend to have a higher recurrence rate [9,12]. The carcinomatous component can occupy 50% to 100% of the tumor [9,43]. The PA areas in CEPA can be challenging to detect, sometimes represented by few epithelial elements or a chondromyxoid stroma. Occasionally, only hyalinized nodules from the remaining pleomorphic adenoma are observed [7]. If the malignant component grows to occupy the entire tumor, detecting PA relies on clinicopathologic data.

In PA, malignancy criteria include pleomorphic and hyperchromatic nuclei, intense mitotic activity, atypical mitoses, necrosis, stromal hyalinization (especially if calcified), capsule invasion, perineural and perivascular invasion, and hemorrhage [9]. Immunohistochemistry can aid in diagnosis [44,45], with intense and diffuse androgen receptor, p53, and HER-2/neu immunoreactivity reported in CEPA [2]. GFAP expression is more common in PA [52,53]. A high Ki67 index (>5%), increased EGFR expression, and loss of Bcl-2 immunopositivity support a diagnosis of malignancy [54,55,56]. Genetic alterations are also present, with PLAG1 and HMGA2 gene fusions/amplification described in both CEPA and PA. TP53 mutations are noticed in 60% of CEPA cases, along with additional mutations, like c-MYC, RAS, P21, and PIK3R1 [56,57,58].

### 2.4. Patterns of Invasion in CEPA

The destructive and infiltrative growth pattern is an important diagnostic criterion [53]. When analyzing a CEPA case, particular attention should be given to the degree of tumor invasion. The term “non-invasive carcinoma developed in a PA” was introduced by LiVolsi and Perzin in 1977 when they described 47 CEPA cases, of which six showed no invasion and did not relapse or metastasize [2,13,59].

Non-invasive carcinomas that develop in PA are classified as in situ or intracapsular. Histologically, these types are characterized by the abrupt transition from typical PA to an area with malignant cytological changes but limited to the pre-existing adenoma. Non-invasive carcinomas can show PA matrix invasion without extracapsular invasion, and these aspects are not considered clinically invasive if the capsule is not penetrated. Microscopically, a clear contrast can be noted between areas of PA replaced by larger cells with pleomorphic and hyperchromatic nuclei and increased mitotic activity and the benign cells of PA. Intense mitotic activity and tumor aspects in the carcinomatous component can be highlighted using markers, such as Ki-67, Her-2/neu, p53, and the androgen receptor, but results must be interpreted with caution as 5–10% of PA may also express these markers [2,7].

The meaning of the term “minimal capsular invasion” has changed over time. Originally reported in 1984 by Tortoledo et al., their study evaluated 40 cases of CEPA, where the majority had tumors with malignant areas showing invasion less than 8 mm from the capsule. The follow-up reported favorable outcomes, with none of these patients dying due to disease progression. However, sixteen patients with invasion exceeding 8 mm died of the disease. The frequency of tumor recurrences was related to the extent of invasion and the status of the resection margins [13,60].

Minimally invasive CEPA is defined as carcinoma with invasion not exceeding more than 1.5 mm of the tumor capsule, though distinguishing between benign tumor pseudopods and true invasion can sometimes be difficult. Mushroom-type invasion into and through the tumor capsule—similar to that described for thyroid carcinomas—is considered indicative of malignancy. For a CEPA diagnosis, histopathologic features of atypical cells always accompany invasion. Atypical changes within these tumors vary from focal to diffuse, with multifocal carcinomatous areas frequently developing and replacing many benign elements. Although some tumors may demonstrate minimal atypia, criteria, such as nuclear hyperchromia and pleomorphism, are common, necrosis is often present, and mitoses are easily observed [2,7].

The cut-off value for CEPA to be considered invasive is still debated, ranging from 1.5 to 100 mm [9,13,28,43,61,62,63]. Early studies reported lesser values for invasion. Brandwein et al. studied 12 patients with non-invasive and minimally invasive CEPA (≤1.5 mm), finding that 8 of these patients did not experience recurrences or metastases 2.5 years after diagnosis [13,61]. Lewis and Olsen reported that for 66 CEPA cases studied, patients with extracapsular invasion of less than 5 mm did not experience tumor recurrences or metastases, but survival rates decreased drastically for those with invasion of at least 15 mm. They concluded that patients with tumors with extracapsular invasion of less than 5 mm had favorable outcomes and responded well to surgical treatment [43]. Based on these studies, CEPAs were classified as non-invasive (in situ, intracapsular), minimally invasive (≤1.5 mm), and invasive (>1.5 mm) [2,7,43]. However, some studies showed that the area of extracapsular extension might be 4–6 mm and even up to 100 mm [9,28].

Silvana Di Palma’s study proposed classifying CEPA into two categories: early CEPA and widely invasive CEPA. Early CEPA is further subclassified into non-invasive, in situ, intratubular, intraductal tumors, and invasive variants represented by intracapsular invasive type, minimally invasive tumors (less than 1.5 mm), and those with invasion not exceeding 6 mm. This proposal is based on recent studies analyzing CEPA with invasion exceeding 1.5 mm, which reported a significant proportion of cases where invasion did not exceed 6 mm and the patients did not show disease progression [13]. The prognostic implications of this classification need further investigation.

### 2.5. The Evolutionary Course of CEPA

Depending on the histopathologic type and the level of invasion, the 5-year survival rate for patients with CEPA can vary widely, from over 90% to just over 20%. Patients with non-invasive or minimally invasive CEPA generally have a prognosis similar to PA, with metastases being rarely reported. In contrast, invasive CEPA is considered extremely aggressive, with 23–50% of patients developing one or more recurrences. Tumor recurrences and metastases are indicators of a poor prognosis, with metastases observed in 30–70% of cases, commonly found in the lungs, bone, and kidney. In these cases, the 5-year survival rate is approximately 50%. CEPA typically metastasizes as carcinoma, although there are reports of metastases with histological features of PA [6,7,64,65,66,67,68].

Non-invasive CEPA metastases are reported in fewer than 2% of cases; thus, non-invasive and minimally invasive CEPA generally have an excellent prognosis, whereas invasive CEPA has a poor prognosis [7]. Non-invasive carcinomas with negative resection margins do not require adjuvant treatments after surgical excision. Minimally invasive carcinomas are treated based on the resection margin status and the presence of perineural invasion—if either is observed, additional surgery and/or radiotherapy is required [2].

Invasive CEPA often includes highly malignant components, such as SDC, ADK NOS, or UC [46]. The 5-year survival rate for CEPA can vary between 30% and 96% depending on the histological type of the malignant component. Specifically, 5-year survival rates are 62% for SDC, 50% for myoepithelial carcinoma, and 30% for UCa as malignant components of CEPA [68,69]. In a series of CEPAs with sinonasal localization and a malignant component of adenoid cystic carcinoma, the overall survival rate was 7.7 years, with a 5-year survival rate of 50% and a 10-year survival rate of 29% [47].

## 3. Clinico-Morphological Aspects in CS

CS cases represent 0.04–0.16% of all malignant salivary gland tumors with distant metastasis occurring in 54% of patients. The CS metastasis is usually described in the lung, but liver, bone, abdominal cavity, and brain metastases are being reported. Patients are in their sixth and seventh decade of life, and mostly there is no gender predilection reported [69,70]. Clinically, patients typically present with a rapidly growing tumoral mass that may be accompanied by pain and facial nerve palsy. Tumors can be well or poorly circumscribed [2,7,25].

### 3.1. Gross and Microscopic Aspects

Salivary carcinosarcomas are biphasic tumors, composed mostly of carcinomatous and sarcomatous elements in variable proportions (Figure 5).

Chondrosarcoma and osteosarcoma are the most common sarcomatous elements of CS. Fibrosarcoma, myxosarcoma, leiomyosarcoma, liposarcoma, rhabdomyosarcoma, and even malignant fibrous histiocytoma are less frequently observed as components of CS. The carcinomatous components of CS are mainly represented by a differentiated/moderately differentiated SDC, UC, adenocarcinomas, or squamous cell carcinoma. Small cell carcinoma, EMC, myoepithelial carcinoma, or ACC were reported in isolated cases as CS components. Invasion and local destruction are characteristics of this neoplasm [3,7,25,26,71,72,73,74].

In one third of the cases, the benign PA area can also be noted. Petersson F and Loh KS described a rare case of CS developed on a PA for which the carcinomatous component was represented by large cells of neuroendocrine carcinoma and a sarcomatous one based on a spindle cell sarcoma with myofibroblastic differentiation [24]. Geraldes Filho et al. described a case of salivary CS with features represented by undifferentiated carcinoma and the mesenchymal component with chondrosarcoma, high-grade undifferentiated sarcoma, and malignant giant cell tumor aspects [73].

### 3.2. The Evolutionary Course of CS

Salivary CSs are aggressive tumors with an average survival rate of 29.3 months, with most patients (60%) dying within 30 months of initial diagnosis. Median survival rates of 3.6 years have also been reported, but the 5-year survival rate is 0%. CS presents local recurrences and metastases. However, due to their rarity, the long-term prognosis is difficult to predict [3,7,68,69,70,71,72,73,74].

There is still no standardized treatment for salivary CS, but the most recent data indicate that the therapy of these tumors is represented by wide surgical excision supplemented by radiotherapy and in selected cases by chemotherapy [74]. Future research should focus on pathogenetically oriented therapies for rarely diagnosed or insufficiently studied diseases [69,70,75,76,77].

Due to their rarity and the heterogeneous morphological aspects, the diagnosis of CS is challenging. The differential diagnosis of salivary CS must include the sarcomatoid variant of a SDC and synovial sarcoma. In SDC, the epithelial component is similar to breast carcinoma, and the sarcomatoid component is immunopositive (but not always) for cytokeratins. Synovial sarcoma resembles a salivary CS both morphologically and in the immunohistochemical profile, but in synovial sarcoma, compact bundles can be identified intersecting spindle cells and glandular structures, while the cells that make up CS are more pleomorphic and poorly differentiated [26,78]. Collision, hybrid, and dedifferentiated tumors must also be excluded. Also, it is not clear if the carcinomatous and sarcomatous elements occur through the collision of two tumors or if these two components are of a clonal origin. The study of genomic profiles of the epithelial and mesenchymal components of the CSs showed an overall homology of 75% between their profiles, which was considered indicative of a monoclonal origin [79]. However there are hypotheses suggesting that CSs are multiclonal and derived from two or more types of stem cells [70]. Further studies are needed to investigate the phenotypes of the components of CS.

## 4. Clinico-Morphological Aspects of MPA

MPA is described in the literature data mostly in female patients, in their third and sixth decades of life. The primary PA is in the parotid gland (79% of cases), submandibular gland (13%), and palate (9%) [34]. MPA is histologically and molecularly identical with the primary PA. Both the PA and its metastasis consist of a mixture of benign epithelial, myoepithelial, and mesenchymal components. The tumors are well circumscribed (both primary and secondary) [29,80]. Most studies show that the histology of the PA is not important for the metastatic ability of this type of tumor. In some cases, pleomorphic cells with mitotic activity were identified. However, these features are not considered sufficient proof for the PA to be classified as a malignant tumor. There are authors who believe that the presence of mitotic activity and the absence of the capsule are aspects indicative of the metastatic potential of PA [81]. Some studies have shown that in MPAs, the stromal or myoepithelial components form the bulk of the tumor mass, and the cells have a high rate of mitoses [6]. A case with local recurrence of PA and simultaneous MPA in 58/59 ipsilateral cervical lymph nodes was reported [33]. Cervical lymph node MPA and infiltration of the sternocleidomastoid muscle were identified by Catarzi et al. [35]. There are reports of patients that died of disease [29]

### The Evolutionary Course of MPA

With a benign histological appearance but malignant behavior, MPA represents a rare aspect of salivary neoplasms. The malignant behavior of these tumors is often associated with multiple local recurrences of PA following surgical interventions or incomplete excision of the primary tumor. Surgical manipulation is believed to lead to the intravascular implantation of tumor cells. Additionally, radiation-induced malignant transformation or metastasis due to investigative maneuvers, such as fine-needle aspiration, are risk factors for developing MPA after the primary tumor has been excised [6,31].

A long interval (1.5–55 years) between the development of the primary tumor and its metastases has been reported [31]. Half of these tumors metastasize to bone, and 30% metastasize to the lungs and neck lymph nodes. Less frequently, MPAs are identified in other parts of the body [4,7,32,69]. In the current literature data, there have been MPAs with localization, such as intra-abdominal (in the liver, kidneys, retroperitoneal), skull, central nervous system, pharynx, in the paranasal sinuses, external auditory canal, larynx, and even in the skin (of the gluteal region). Some authors reported that the metastasis developed on average 15–16 years after the initial PA diagnosis. A case with PA of the minor salivary glands of the palate vault untreated for 40 years developed an MPA in the cervical lymph nodes [82]. There are other studies that have identified situations where PAs have metastasized 51–55 years after the primary diagnosis [4,32,81,82,83,84].

The prevention and treatment options for MPA remain subjects of ongoing debate. Complete tumor resection with adequate margins can prevent local recurrence and distant metastasis. For the minor salivary gland, the reported recurrence rate of PA is low (with this, the incidence of MPA from this location also decreases) [85]. Therapeutic agents targeting progesterone receptor molecular signals are considered potential candidates for treating recurrent PA [31,81]. The treatment of choice for MPA is the surgical excision of the metastasis with wide tissue margins. In some cases, postoperative radiotherapy and chemotherapy were used as adjunctive treatments with variable results [82].

Ongoing documentation and long-term follow-up are mandatory so a survival rate for MPA can be assessed. According to the literature data, about 40% of MPA patients die with disease [82].

## 5. Conclusions

CEAP, MPA, and CS represent mixed salivary gland tumors with aggressive behavior. Different tumors can develop in a CEPA, so treatments must take into account the histological type of carcinoma developed in the PA. The pathologist must look at these CEPAs not as a unique type of tumor but as a group of distinct tumors, for which the gross and microscopic appearance must be adequately evaluated. The pathologist’s report for CEPA cases must include, in addition to the type of tumor, the presence and degree of invasion, as these are important aspects for prognosis and therapy. CEPA with malignant components represented by UC, SDC, and ACC or with high-grade transformation areas usually receive a more complex treatment. The subtype of CEPA with extensive extracapsular invasion is aggressive and has an increased risk of recurrence and metastasis. In practice, the extent of invasion in the cases of CEPA and the identification of the benign component may be challenging. Even though PAs are common tumors of the salivary glands, their metastatic potential has to be acknowledged. Understanding the tumorigenesis of MPAs may help in the prevention of the malignant behavior of the PA. In cases with CS, recognizing the biphasic morphology is of paramount importance in the differential diagnosis. The mesenchymal PA component of the CEAP has to be differentiated from the malignant component of CS. CSs are aggressive tumors that have a poor prognosis.

## Figures and Tables

**Figure 1 diagnostics-14-01942-f001:**
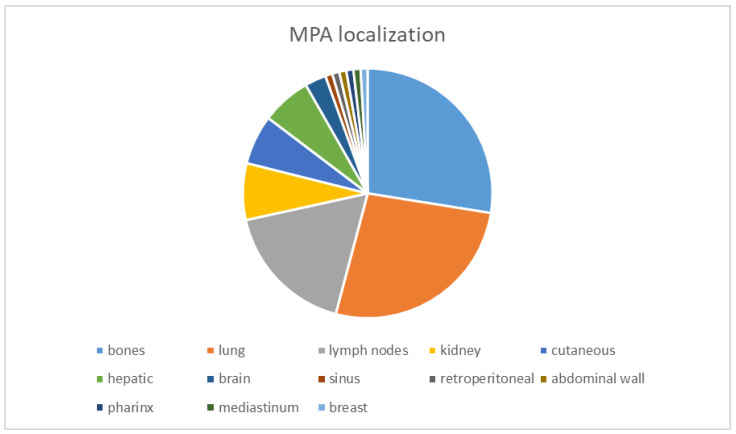
Distribution of MPA localization; from literature-data-counted cases, the MPAs identified were 28% in bone (30 cases), 29% in the lung (29 cases), 17% in lymph nodes (19 cases), 7% in the kidney (8 cases), 6% cutaneous and hepatic (each with 7 cases cited), and 3% in the brain (3 cases identified), and the rest represents 1% being isolated cases [4,30,31,32,33,34,35,36].

**Figure 2 diagnostics-14-01942-f002:**
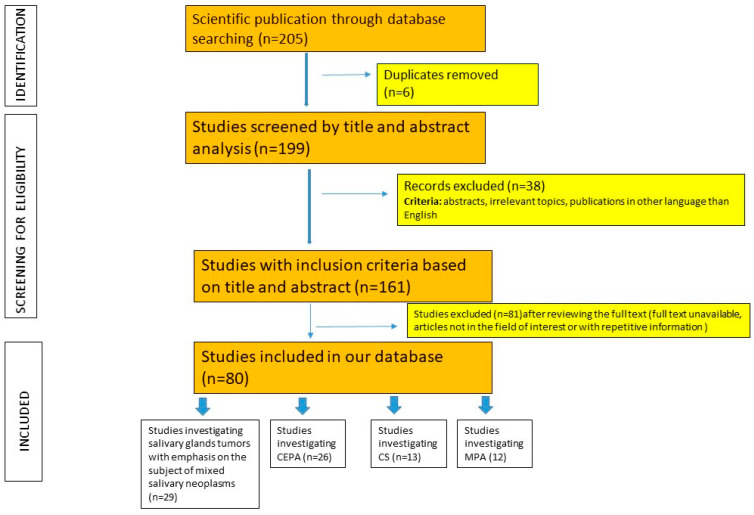
Algorithm used for article selection; n = number of studies; CEPA = carcinoma ex pleomorphic adenoma, CS = carcinosarcoma, MPA = metastasizing PA.

**Figure 3 diagnostics-14-01942-f003:**
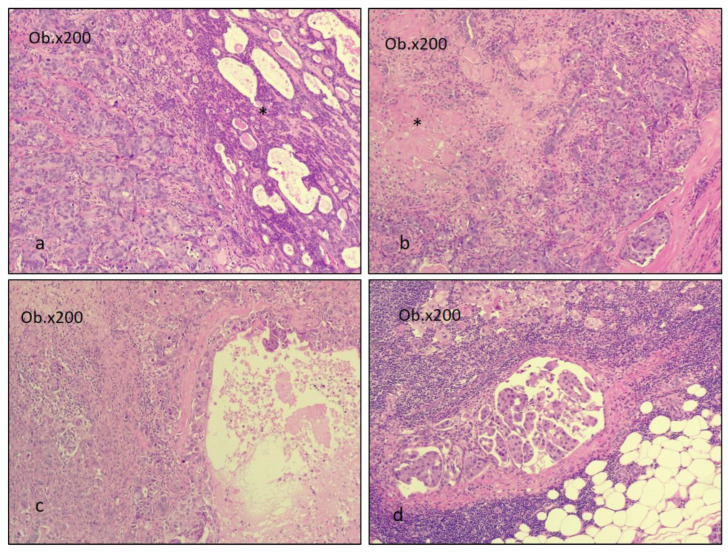
SDC as the malignant component of CEAP. The benign PA is presented in images (**a**,**b**) and is marked with *. Image (**c**) is only the SDC component, and (**d**) illustrates the metastasis in the lymph node of the SDC malignant component of CEAP. The images were obtained using a Leica DM750 microscope with a digital camera, 200× magnification (Ob. = objective).

**Figure 4 diagnostics-14-01942-f004:**
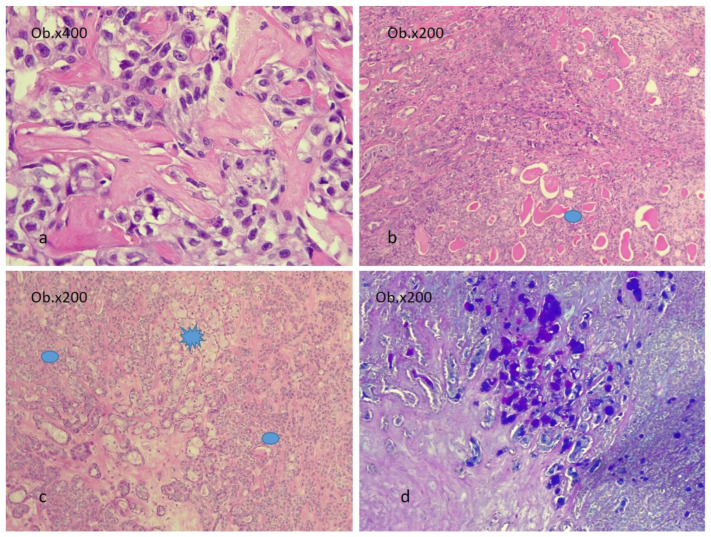
CEPA. (**a**). Benign component of PA with myoepithelial cells. (**b**). ADK NOS as the malignant component with PA marked with a blue oval shape. In images (**c**,**d**), MEC is the malignant component marked with a blue star and PA is marked with oval shapes. In image (**d**), the Alcian Blue histochemical stain highlights the MEC component. The images were obtained using a Leica DM750 microscope with a digital camera, 200× and 400× magnification (Ob. = objective).

**Figure 5 diagnostics-14-01942-f005:**
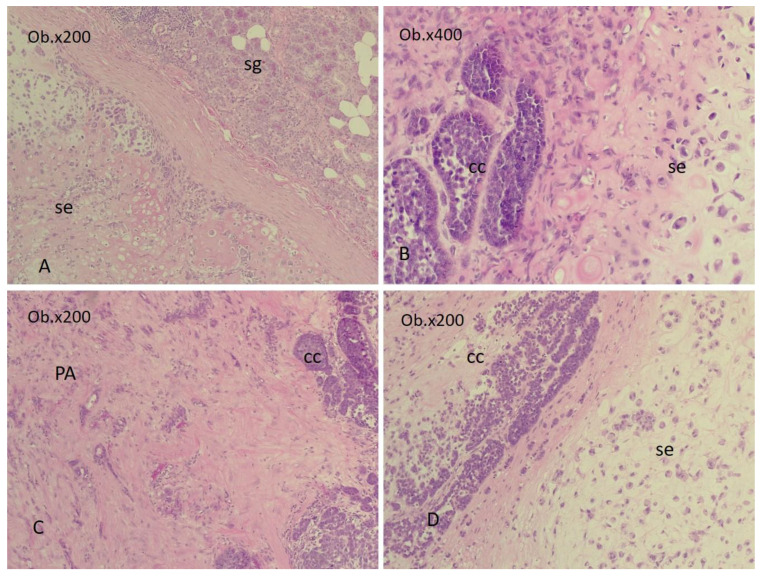
CS with PA, sarcomatous elements (se) of chondrosarcoma, and basal cell adenocarcinoma (BC ADK) as the carcinomatous element (cc). (**A**). CS-condrosarcoma (se) and adjacent salivary gland (sg). (**B**). CS with BC ADK (cc) and condrosarcoma (se) areas. (**C**). CS with PA and BC ADK (cc) areas. (**D**). CS with BC ADK (cc) and condrosarcoma (se) areas (lower magnification). Abbreviations: sg = salivary gland, se = sarcomatous element, cc = carcinomatous element, PA = pleomorphic adenoma. The images were obtained using a Leica DM750 microscope with a digital camera, 200× magnification (Ob. = objective).

## Data Availability

No new data created.

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
