# Peer review of "Clinical and Morphological Aspects of Aggressive Salivary Gland Mixed Tumors: A Narrative Review"

_diagnostics, 2024, doi:10.3390/diagnostics14171942_

Round 1
Reviewer 1 Report
Comments and Suggestions for Authors
The review is a comprehensive summary of the current understanding of salivary glands mixed tumors, which may serve as a good reference source for related studies. The manuscript is well prepared. However, the following points need to be addressed.
1. It would be better to show the percentage of each site in the figure1
2. The review needs to be thoroughly edited to include more precise information and to cite more relevant studies.
3. Is there an expert consensus on the definition and classification of the salivary glands mixed tumors?
4. The review focused on clinical and morphological aspects. Are there any studies on the correlation between molecular alterations and histopathological changes?
Author Response
Dear Reviewer,
Thank you for taking the time to evaluate our manuscript and for your feedback. Your input has enhanced the clarity and accuracy of our paper, and we greatly value your expertise.
- At your suggestion we have show the percentage of each site at the explanatory notes in the figure 1 leggend. We would have been put the percentage on the image itself but we found that because of the areas with only one percentage the image wolud resulted crowded as appearance.
- We have edited the review, rephrase the sentences, introduced new informations and referrences as you suggested.
- For your no 3 question-yes, there is a consensous –the names of the mixed tumors are as presented in the text of our research (pleomorphic adenoma , carcinoma ex pleomorphic adenoma , carcinosarcoma and metastasising pleomorphic adenoma) and also in WHO classification. We refere you to consult a study in which is presented the evolution of terms in WHO clasification over the years: Michał Żurek, Łukasz Fus, Kazimierz Niemczyk & Anna Rzepakowska Salivary gland pathologies: evolution in classification and association with unique genetic alterations, European Archives of Oto-Rhino-Laryngology , Volume 280, pages 4739–4750, 2023. This research also provides information about the 4th question
- Yes there are studies correlating molecular alteration and histopathological changes-from the perspective of which alteration are presented in PA and CEPAs. We have referred at these molecular alteration in our text briefly. We did not have more comments about this aspects because in the CS and MPA cases, there are only isolated studies that mentioned this subject. This situation is due to their rarity. So, we feel that conclusions can not be made if the information in reduced to only a few cases. However our research was concentrated on the clinical and morphological aspects that can offer a support for diagnosis and treatment of such tumors. We hope that you will accept our point of view.
- The new text, the revised area are highlighted in red colour. We also added the algorithm used for the selection of the scientific papers for our study.
Anyway, we would like to thank you for your time and for us your revision was helpful. Best regards. The authors.
Reviewer 2 Report
Comments and Suggestions for Authors
Dear Authors,
Please, adapt your review point-by-point according to PRISMA checklist for systematic or scoping reviews. The addressed questions are also missing from the end of the introduction. Please, formulate your questions according to PICO questions for research.
The content and language are fine, just needs to be properly structured!
Author Response
Dear Reviewer ,
thank you for your input. At your suggestion, we have provided the research purpose in the introduction section and the algorithm. We would like to explain our choice of not introduce these section before. We have chosen to present the results of our study in the form of a narrative review because the scientific papers presenting salivary gland neoplastic lesions are few because of their rarity. Also, most of the studies present only isolated cases which makes difficult to have a broad aspect of these rare neoplasms. If you follow the topic like we are you can notice that the reported studies are growing in numbers but this happens slowly. We can tell you that in our center of diagnosis, about 300 tumors were identified in the last 17 years but the majority are pleomorphic adenomas or Warthin tumors. By studding the literature data we notice that the algorithms that you suggested are more commonly used in the systematic reviews combined with meta-analysis and we only offered a narrative one. However, we like how the study looks now-we think that it was a good suggestion. We refined the text, rephrase sections and introduced new data. The new text, the revised area are highlighted in red colour. We also added the algorithm used for the selection of the scientific papers for our study.
Best regards, the authors.
Reviewer 3 Report
Comments and Suggestions for Authors
Interesting and well written article.
I would suggest Authors to expand discussion regarding minor salivary gland tumors: to this aim, Authors could add some references from the recent literature, such as
Oral and oropharyngeal malignant minor salivary gland tumors: A retrospective study. J Stomatol Oral Maxillofac Surg. 2024 Apr 25:101893.
Author Response
Dear reviewer, thank you for your input.
We have reorganized the document and added new data about the tumors with location in the minor salivary glands. Due to your request to investigate more the field of diagnosis of minor salivary gland neoplasia we have enriched our review with new data. Thank you for that. Unfortunately, because of the rarity of these tumors, the studies are few and the ones reporting tumors of the minor salivary glands are even fewer. We hope that with the new data and the revision of the narrative review that we have made, things are now more complete. The new text, the revised area are highlighted in red colour. We also added the algorithm used for the selection of the scientific papers for our study.
Thank you! Best regards. The authors.
Reviewer 4 Report
Comments and Suggestions for Authors
The presented article describes an aggressive salivary gland mixed tumor with clinical and histopathologic features. Unfortunately, the article does not meet the criteria for a review. There is no aim of this review. There is no description of the literature search methodology. Which databases were searched? What keywords were used in the search? What was the time range for the added literature, last 5 years, 10 years or more? What were the inclusion and exclusion criteria for the search? The article has no logical structure. The description of the figures is very laconic. Are the photographs of histologic specimens from the center presenting the work or are they taken from the literature? If so, permission to copy the figures is required.
Writing about prognosis without information about treatment methods is completely useless. Many factors other than histopathologic aspects determine the prognosis of patients. The article adds nothing to the scientific literature. Article is not suitable for publication.
Comments on the Quality of English LanguageModerate editing of English language required
Author Response
Dear Reviewer,
Thank you for taking the time to evaluate our manuscript and for your feedback. Your input has enhanced the clarity and accuracy of our paper, and we greatly value your expertise.
We have rephrased the text, and added the algorithm used and the databases. We will address all of the comments:
- Our review is a narrative one. We have chosen this form of presenting our work because of the rarity of the reports covering the subjects of the salivary gland neoplasms. There are many examples in the literature in which a narrative review presents a topic without a PRISMA algorithm. We feel that it can be presented as such. We have published other reviews in narrative form quite successfully. We suggest that you consult scientific papers like for example: Parums DV. Editorial: Review Articles, Systematic Reviews, Meta-Analysis, and the Updated Preferred Reporting Items for Systematic Reviews and Meta-Analyses (PRISMA) 2020 Guidelines. Med Sci Monit. 2021 Aug 23;27:e934475. doi: 10.12659/MSM.934475. PMID: 34421116; PMCID: PMC8394590. There are opinions and opinions but still we ask you to consider ours also.
- We truly believe that you cannot chose a limited period in time for the salivary glands tumors-a 5 or 10 year as you have written for a review because there are rarely reported tumors. You will end up with a small sample and just few ideas. Most of the studies report a 5% incidence of the salivary glands tumors in the head and neck region and the majority are pleomorphic adenomas. It is a field in development and we have to develop whit it. You have to know what the evolution in terms was over the years to understand how the domain had changed from the first publication of the WHO blue book. We had done that over the years. So, we did not put a time line for the research as you can see in the reference list. We hand-searched all the scientific papers that had as a subject mixed tumors of the salivary glands. You have the algorithm in the new revised manuscript.
- We have added more information in the description of figures. We are sorry for the lack of information regarding copyright, we did not feel that we have to write in the text that the figures are original. They are original and the journal will for sure ask us if they are our own work.
- Our researched was centred on clinical aspects and morphology but we had also written about the treatment. We have refreshed existing data and added new data. Writing about prognosis in relation with morphology and agressive behavior of the tumors is relevant, in our opinion. There are studies confirming that and you can find them in our reference list. Salivary gland tumors are rare, the MPA and CS cases are even more rare, so the histopathologic aspects are important in determination of the prognosis.
Once again, we thank you so much.
Best regards,
The authors.